# Development by Mechanochemistry of La_0.8_Sr_0.2_Ga_0.8_Mg_0.2_O_2.8_ Electrolyte for SOFCs

**DOI:** 10.3390/ma13061366

**Published:** 2020-03-18

**Authors:** Francisco J. Garcia-Garcia, Yunqing Tang, Francisco J. Gotor, María J. Sayagués

**Affiliations:** 1Departamento de Ingeniería y Ciencia de los Materiales y del Transporte, Universidad de Sevilla, E-41092 Seville, Spain; 2Instituto de Investigación en Energía de Cataluña, Sant Adrià de Besòs, E-08930 Barcelona, Spain; yqing518@gmail.com; 3Instituto de Ciencia de Materiales de Sevilla, Centro mixto CSIC-US, E-41092 Seville, Spain; fgotor@cica.es (F.J.G.); sayagues@cica.es (M.J.S.)

**Keywords:** SOFC, mechanochemistry, LSGM, electrolyte, perovskite, ionic conductivity

## Abstract

In this work, a mechanochemical process using high-energy milling conditions was employed to synthesize La_0.8_Sr_0.2_Ga_0.8_Mg_0.2_O_3-δ_ (LSGM) powders from the corresponding stoichiometric amounts of La_2_O_3_, SrO, Ga_2_O_3_, and MgO in a short time. After 60 min of milling, the desired final product was obtained without the need for any subsequent annealing treatment. A half solid oxide fuel cell (SOFC) was then developed using LSGM as an electrolyte and La_0.8_Sr_0.2_MnO_3_ (LSM) as an electrode, both obtained by mechanochemistry. The characterization by X-ray diffraction of as-prepared powders showed that LSGM and LSM present a perovskite structure and pseudo-cubic symmetry. The thermal and chemical stability between the electrolyte (LSGM) and the electrode (LSM) were analyzed by dynamic X-ray diffraction as a function of temperature. The electrolyte (LSGM) is thermally stable up to 800 and from 900 °C, where the secondary phases of LaSrGa_3_O_7_ and LaSrGaO_4_ appear. The best sintering temperature for the electrolyte is 1400 °C, since at this temperature, LaSrGaO_4_ disappears and the percentage of LaSrGa_3_O_7_ is minimized_._ The electrolyte is chemically compatible with the electrode up to 800 °C. The powder sample of the electrolyte (LSGM) at 1400 °C observed by HRTEM indicates that the cubic symmetry Pm-3m is preserved. The SOFC was constructed using the brush-painting technique; the electrode–electrolyte interface characterized by SEM presented good adhesion at 800 °C. The electrical properties of the electrolyte and the half-cell were analyzed by complex impedance spectroscopy. It was found that LSGM is a good candidate to be used as an electrolyte in SOFC, with an Ea value of 0.9 eV, and the LSM sample is a good candidate to be used as cathode.

## 1. Introduction

The solid oxide fuel cells (SOFCs) are one of the cleanest sources of energy that transform chemical into electrical energy and are proposed as an alternative to the use of fossil fuels. SOFCs have a high conversion efficiency and can operate with various fuels (hydrocarbons, ethanol, natural gas, H_2_, etc.). In the last decade, the number of investigations focused on optimizing the components for these SOFCs has increased considerably [1,2,3,4,5,6,7,8]. However, from an economic point of view they are still not commercially viable. Electrolytes are a key component of SOFCs, because in many cases they also act as support for the cell. The selection of materials as electrolytes is of great importance for the development of SOFCs. It is necessary to take into account a large number of parameters, (1) the ionic conductivity that must be high and stable over time; (2) the chemical stability and compatibility with the electrodes (cathode and anode); (3) the sinterability, which should allow a high densification, as it acts as a physical barrier to prevent the mixing of fuel and oxidant gas streams; (4) the thermal expansion coefficient that must be similar to that of the electrodes to avoid mechanical failures; and (5) the cost and ease of processing.

Although the most commonly used electrolyte materials are based on ZrO_2_ and CeO_2_, in recent years a large number of investigations on La_1−x_Sr_x_Ga_1−y_Mg_y_O_3−δ_ (LSGM)-mixed oxides with perovskite structures are being carried out [9,10,11,12,13,14,15,16,17,18]. The replacement of La^3+^ by Sr^2+^ and Ga^3+^ by Mg^2+^ generates oxygen vacancies in the structure, enhancing the oxygen ion conductivity. However, it must also be taken into account that the cation substitution could modify the structural symmetry, or even induce the appearance of some secondary phases [19,20,21,22,23]. LaGaO_3_ presents an orthorhombic structure (Pbnm SG) at room temperature; however, La_1−x_Sr_x_Ga_1−y_Mg_y_O_3−δ_ shows orthorhombic, rhombohedral, or cubic symmetry, depending on the total amount of dopants (*x + y*). At (*x + y*) ≤ 0.25, the material shows orthorhombic symmetry, at 0.25 < (*x + y*) ≤ 0.30, a mixture of orthorhombic and rhombohedral symmetry (R-3c SG) exists, and at (*x + y*) ≥ 0.35 with either *x* or *y* ≥ 0.20, a cubic symmetry (Pm-3m SG) is observed. Nevertheless, the situation at the SOFC operating temperature may be different [22], since a change in the structure may also occur with increasing temperature.

The specific free volume of the unit cell, as defined by Sammells et al. [24], together with the lattice stability, determined by the tolerant factor, are the two main factors that influence the ionic conductivity in perovskite-type oxides. Hayashi et al. [25] carried out electrolyte conductivity studies in a double-doped perovskite structure, finding that there is a compromise between the tolerance factor and the specific free volume. There is an almost linear correlation between the specific free volume of the structure and the (*x* + *y*) concentration [26], which is directly related to the ionic conductivity of the material. Increasing (*x* + *y*) rises the ionic conductivity, however, the solubility limit for Sr and Mg in the La_1−x_Sr_x_Ga_1−y_Mg_y_O_3−δ_ solid solution is *x* = *y* = 0.2.

It has been shown in previous studies that mechanochemistry is an appropriate method to synthesize complex solid solutions with a perovskite structure that can be used as an electrolyte [27,28] (LSGM system), cathode [29,30] (LSM system, La_1−x_Sr_x_MnO_3_), or anode [31] (SLT system, Sr_1−x_La_x_TiO_3_) in SOFCs. Mechanochemistry is a relatively simple process that uses the mechanical energy provided by ball mills to induce solid-state reactions at room temperature in a wide variety of powder reactant mixtures. One of the most important advantages of mechanochemistry is its ability to produce large material quantities with the desired stoichiometry without any additional annealing process, which allows for significant energy savings. Gonçalves et al [27] used the mechanochemistry to synthesize the LSGM system; however, they did not obtain the pure perovskite structure after 6 h of milling. Moure et al [28] obtained LaGaO_3_ also by mechanochemistry, but after 36 h of milling. Long-term mechanochemical processes can induce a high level of contamination from the milling media in the final product, which can be detrimental to the conductivity properties. The possibility of transferring increasing amounts of mechanical energy using high-energy ball milling equipment, such-as planetary mills, means that milling times can be shortened considerably, reducing the impurity content in the final product to admissible levels for a large number of applications. In this sense, in previous works [29,30], our research group synthesized the LSM system in just 90 min using high-energy milling conditions, demonstrating that mechanochemistry can be direct, cost-effective, time-saving, and suitable for large scale production.

In the present work, a similar procedure was applied to obtain the LSGM system even in a shorter time. Besides, a La_0.8_Sr_0.2_MnO_3_//La_0.8_Sr_0.2_Ga_0.8_Mg_0.2_O_2.8_ (LSM//LSGM) half-cell was built using both systems with the same perovskite structure and synthesized from the same mechanochemical procedure, with the aim of avoiding chemical reactivity, getting good compatibility between electrolyte and electrode, and preserving the structural integrity of the cell. Both the electrolyte and the electrode were fully characterized from a structural, chemical and electrical point of view. The chemical compatibility between both components was also studied and the efficiency of the half-cell was measured.

## 2. Materials and Methods 

Powders of LSGM and LSM were prepared using a mechanochemical method from the stoichiometric amounts of the corresponding cation oxides, according to the Equations (1) and (2), respectively:0.4 La_2_O_3_ + 0.2 SrO + 0.4 Ga_2_O_3_ + 0.2 MgO → La_0.8_Sr_0.2_Ga_0.8_Mg_0.2_O_2.8_(1)
0.4 La_2_O_3_ + 0.2 SrO + 0.5 Mn_2_O_3_ → La_0.8_Sr_0.2_MnO_3_(2)

Lanthanum oxide (Fluka, Bucharest, Romania, 99.98% in purity), strontium carbonate (Panreac, Barcelona, Spain, 98% in purity), gallium oxide (Strem Chemicals, Newburyport, MA, USA, 99.998% in purity), magnesium oxide (Strem Chemicals, 99.5% in purity), and manganese oxide (III) (Aldrich Sigma, Madrid, Spain, 99% in purity) powders were used as the starting reactants. Due to the unviability of using SrCO_3_ as raw material in the mechanochemical process, it was first heated in a furnace at 1100 °C for 4 h to obtain SrO. The stoichiometric amount of SrO was weighed immediately after annealing, and the milling process was started quickly to avoid errors by the fast re-carbonation. The rest of the oxides were also heated to the same temperature to eliminate the corresponding hydration products or hydroxides. 

For the mechanochemical experiments, seven tungsten carbide (WC) balls and stoichiometric quantities of the powder reactants to produce 3 g of the perovskite sample were placed in a hardened chromium steel jar and milled in a planetary ball mill (model Micro-Mill Pulverisette 7, Fritsch, Idar-Oberstein, Germany) at a spinning rate of 600 rpm. The volume of the vial was 60 mL. The diameter and weight of the balls were 15 mm and 26.4 g, respectively. The powder to-ball mass ratio (PBR) was approximately 1:62. To avoid contamination of the obtained powders (with Fe or WC), a first batch of sample was prepared and rejected, so that for the following batches the walls of the jar and the balls were protected with the desired perovskite composition.

The X-ray diffraction (XRD) patterns were obtained using a PANalytical X’Pert Pro model diffractometer (Malvern PANalytical Ltd., Almelo, The Netherlands) with Bragg-Brentano geometry and an X’Celerator detector. The diffraction patterns were scanned from 10 to 80° (2θ) in step-scan mode at a step of 0.05° and a counting time of 300 s/step. The patterns were corrected with a standard (polycrystalline Si). The quantification of phases detected in the XRD patterns was performed by means of the Rietveld method using the X’Pert HighScore Plus software (version 3.0.5, PANalytical B.V., Almelo, The Netherlands). Background, zero point, scale factor, pseudo-Voigt parameters of the peak shape and cell parameters were refined.

The chemical compatibility of the components with temperature was analyzed by X-ray diffraction in a PANalytical X’Pert-Pro instrument equipped with an “Anton Parr high-temperature attachment (HTK 1200) ” and a Q-goniometer/Q using Cu Kα radiation, a secondary Kβ filter and an X’Celerator detector. The patterns were recorded at atmospheric pressure at temperature intervals of 50 °C from room temperature to 800 °C, the heating rate was 5 °C/min, and the scanning speed was 5.4 °/min.

Microstructural characterization was carried out by scanning and transmission electron microscopy (SEM and TEM) techniques. Powder samples were dispersed in acetone and droplets of the suspension were deposited onto a holey carbon copper grid. The SEM images were obtained on a Hitachi S-4800 SEM-FEG microscope (Hitachi High-Tech, Fukuoka, Japan) in secondary electron mode at an acceleration voltage of 5 kV. Energy-dispersive X-ray analysis (EDX) was performed with a detector coupled to the SEM microscope, using an acceleration voltage of 20 kV. The TEM/HRTEM images were taken on a 300 kV TECNAI FEG microscope (model G2 F30 S-twin, FEI Company, Hillsboro, OR, USA) (0.2 nm point resolution) with scanning-transmission capabilities (STEM). The HR micrograph analysis, lattice spacing, fast Fourier transform (FFT), and phase interpretation was done with the Gatan Digital micrograph software (Gatan AMETEK Inc., version 2, Elancourt, France) and the Java version of the Electron Microscope Software (JEM) (Taipei, Taiwan).

Impedance spectra of LSGM electrolyte and the built LSM//LSGM half-cell were measured using an impedance analyzer model Solartron 1260A (AMETEK Scientific Instruments, Oak Ridge, TN, USA) at open circuit voltage (OCV). All measurements were made from 800 °C to room temperature, with a range of 100 °C. The electrolyte was measured in a synthetic airflow with a frequency range of 1 MHz–0.1 Hz and the LSM//LSGM half-cell was carried out in synthetic airflow and 1 MHz–0.1 Hz frequency range. All the spectra were analyzed with the ZView software (version 3.0, Scribner Associates, Inc., Southern Pines, NC, USA). Pt wires were used as voltage, and current collectors in a single chamber configuration.

For the impedance measurements, 0.6 g of the LSGM powder was compacted in a disk shape (ϕ = 12 mm and thickness = 1 mm) using a uniaxial press at 10 MPa for 5 min and further sintered at 1400 °C in atmospheric air for 10 h, at heating and cooling rates of 5 °C/min, in a tubular furnace. The density of sintered LSGM was measured by the water immersion technique (Archimedes method). Previously to the electrolyte conductivity measurement, platinum paste was deposited, using the brush-painting technique. The deposition was made on both sides with an intermediate heat treatment at 200 °C for 30 min and, finally, for a correct platinum adherence, the electrolyte was heated to 800 °C (heating rate of 5 °C/min) for 1 h.

For the SOFC construction, the electrodes were deposited on both sides of an LSGM electrolyte disk using the same brush-painting technique from a slurry of 25 wt.% LSM, 25 wt.% of LSGM and 50 wt.% terpineol. The electrode films were subsequently heat-treated (800 °C for 4 h) to give consistency and promote their adhesion and densification; to finish, the platinum paste was deposited in the same way above described.

## 3. Results and Discussion

### 3.1. Microstructural Analysis and Thermal Stability

LSGM and LSM samples were synthesized by milling the stoichiometric amounts of the reactants according to Equations (1) and (2) for 60 and 90 min, respectively. The required synthesis time was fixed after checking by XRD at increasing milling time the total conversion of the reactants into products. The desired final product was obtained without the need for any subsequent annealing treatment. Figure 1 shows the XRD patterns of the LSGM electrolyte and LSM cathode after the milling processes. All diffraction maxima of the LSGM (Figure 1a) can be indexed to the perovskite structure with cubic symmetry (space group 221, Pm-3m; JCPDS Card N°: 52-0022). A lattice parameter a = 0.390 nm and a coherent diffraction domain, D = 26 nm, were determined. It is important to take into account that the original perovskite structure (LaGaO_3_) was doped in both A and B positons to generate oxygen vacancies. Sr^2+^ (ionic radius = 1.44 pm) substitutes 20% of La^3+^ (ionic radius = 1.36 pm) ions in the cuboctahedra positions and Mg^2+^ (ionic radius = 1.72 pm) substitutes 20% of Ga^3+^ (ionic radius = 0.62 pm) ions in the octahedral positions. In the case of the LSM sample (Figure 1b), all maxima were also indexed in a pseudo-cubic symmetry with a similar lattice parameter, a = 0.391 nm. It is known from the literature data that the LSM structure is rhombohedral (space group 167, R-3c; JCPDS Card N°: 1-085-2219), however, due to the large peak broadening, because of the small size of the coherent diffraction domains (D = 14 nm), the real symmetry could not be resolved [29,30]. In this case, La^3+^ are substituted by 20% of Sr^2+^ and the B positions are totally occupied by Mn^3+^ (ionic radius = 1.645 pm).

The thermal stability of the electrolyte LSGM was studied between room temperature and 1600 °C (dwelling time of 2 h). Up to 800 °C, only the pure LSGM phase is observed in the XRD patterns (Figure 2a). From 900 °C, a small amount of two secondary phases begins to appear as can be observed in the corresponding XRD patterns (Figure 2b). These phases were identified to be LaSrGa_3_O_7_ (tetragonal structure and P-421m space group; JCPDS Card N°: 45-0637)) and LaSrGaO_4_ (tetragonal structure and I4/mmm space group; JCPDS Card N°: 01-080-1806)). However, from 1400 °C, the secondary phase LaSrGaO_4_ disappears (Figure 2b). The XRD peak broadening decreases as the temperature increases, which consequently implies that the size of the crystalline domains increases [32]. Note that the LSGM phase maintains the cubic symmetry over the whole temperature range studied, in agreement with the doping level used, i.e., (*x + y*) = 0.4.

The percentage of the main phase and the two secondary phases between 900 and 1600 °C was quantified by the Rietveld method using the X’Pert HighScore Plus software, and the results are shown in Table 1. According to published works [33,34,35,36], the existence of the secondary phase LaSrGa_3_O_7_ is a common problem in all synthesis processes. Yamaji et al. [34] reported that Ga_2_O_3_ has a high coefficient of vaporization at high temperature in a reducing atmosphere; therefore, a small amount of Ga leaves the perovskite structure, forming the secondary phase LaSrGa_3_O_7_ by reacting with Sr cation dopant. Nevertheless, the presence of the secondary phases LaSrGa_3_O_7_ and LaSrGaO_4_ was also observed when LSGM was obtained using oxidizing conditions [35,36]. Wu et al. [15] indicated that the material LaSrGa_3_O_7_ is an insulator, negatively affecting the conductivity of the LSGM sample. However, Marrero-López et al. [36] observed that the total conductivity of LSGM is not significantly affected by the presence of LaSrGa_3_O_7_ and LaSrGaO_4_, even for samples containing substantial amounts of impurities. In any case, the sintered temperature chosen to consolidate the LSGM electrolyte was 1400 °C in order to get the minimum quantity of LaSrGa_3_O_7_ (see Table 1), which was in accordance with previous studies [35,36]. The thermal stability of the LSM electrode was analyzed in a previous work [29], finding that the structure is stable until 1100 °C.

The microstructural results obtained by TEM/HRTEM of the LSGM electrolyte powder annealed at 1400 °C are shown in Figure 3. The sample consists of agglomerates formed by particles with sizes of about 200 nm, as can be observed in the TEM images (Figure 3a,b). Figure 3 shows that annealing at 1400 °C induces limited grain growth and a nanometric/submicrometric microstructure is maintained. In the same particle, several misoriented nanodomains were found. Some oriented nanocrystalline domains are shown in the HRTEM micrographs (Figure 3c,f) and different zone axis of the cubic structure (space group Pm-3m) were analyzed ([1 0 1]; [1 1 1] and [2 1 3]), corroborating that the symmetry was maintained after the thermal treatment. The FFT of these nanocrystals (white squares) and the corresponding simulated electron diffraction patterns (EDP) are inset. 

### 3.2. Electrolyte-Electrode Interface

The chemical compatibility between the electrolyte and the electrodes is an important feature to be taken into account, because the electrolyte is in contact with the electrodes at high temperature for a prolonged time, which can promote the diffusion of the ions, reducing the conductivity of the materials. To analyze this compatibility, a dynamic X-ray diffraction study as a function of temperature was carried out. Figure 4 shows the obtained results between 30 and 800 °C for a 1:1 electrolyte-electrode mixture (LSGM-LSM). The maximum temperature of 800 °C was chosen since it is the typical operating temperature of this kind of cells. Although the structure of both phases is similar and difficult to differentiate (note that both compounds have virtually identical lattice parameters), the results show that the only phase present in the mixture heated up to 800 °C is the pseudo-cubic phase with perovskite structure. No other secondary phase was detected. It can be concluded that the electrolyte and the cathode are chemically compatible at least up to 800 °C.

The electrode/electrolyte interface of the half-cell prepared by brush-painting and treated at 800 °C for 4 h was studied by SEM/EDX. The cross section SEM image in Figure 5 revealed a good adhesion between the electrode (upper layer, formed by LSM/LSGM (1:1)) and the electrolyte (LSGM) that will favor the performance and reliability of the cell. The electrode showed an adequate porosity at a microstructural level, which will favor the gas diffusion, and the electrolyte a suitable degree of densification (relative density of 89%), favoring the conduction of oxygen ions. EDX analyses performed exactly at the interface did not evidence any sign of chemical interdiffusion between both components. As an example, two EDX analyses are presented in Figure 5. Note that Mn is only observed in the electrode. The inset in Figure 5 presents the fracture surface of the electrolyte. A transgranular fracture was observed with the presence of particle cleavages, characteristic of a brittle material. A particle size in the range of 2–3 μm can be deduced from the fracture surface of the sintered LSGM. Regions with a darker contrast were associated with the presence of the LaSrGa_3_O_7_ secondary phase.

### 3.3. Electrical Properties and Conductivity Measurements

The impedance spectra of the electrolyte at several temperatures are presented in Figure 6; they are formed by a semicircle that is relative to grain and grain boundary conductions and an arc associated with interface conduction between LSGM and the platinum electrodes. The conductivity relaxation of polycrystalline materials, i.e., LSGM, generally consists of grain (bulk), grain boundary, and specimen-electrodes interface processes from high to low frequencies. Therefore, for well-separated grain, grain boundary and electrode relaxation processes, a complete semicircle corresponding to each process is observed on the impedance spectra [17,18]. The resultant spectra are fitted with an equivalent circuit that consists of a parallel combination of respective resistances and constant phase elements connected in series. The constant phase elements are needed when depressed semicircles are observed in the impedance spectra, as in this work. Variations from a semicircle are associated with local charge inhomogeneity due to surface roughness, the presence of pores and secondary phases, or variations in composition [17,18].

In this study, it was not possible to discern the partial contribution of grains and grain boundaries. This can be in principle due to (i) an incomplete densification of the material or (ii) a very small size of the grains that enhances grain boundary conduction, hindering bulk conduction. The last hypothesis is based on the fact that LSGM possess a relatively high total conductivity, as discussed fully below. Further, it is evident that the bulk conductivity semicircle is shifted towards higher frequencies with the increasing temperature, becoming considerably smaller. In fact, at temperatures higher than 700 °C, the semicircle is almost disappearing, and curved line behavior is observed, which represents specimen-electrode contribution. This yields in a reduction of the total resistance of LSGM with the increasing temperature. The grain and grain boundary resistances are calculated from the intercepts of the semicircular arcs on the real part of the impedance axis (x axis), being the total resistance the sum of the two, excluding the specimen-electrode contribution. 

The total conductivity of LSGM is calculated from the total resistance and the geometric dimension of the pellet (sample thickness and the sample area of the platinum-coated face of the sample). As expected, the conductivity of the LSGM sample increases with the increasing temperature. A conductivity of 0.093 S/cm at 800 °C was obtained for the LSGM electrolyte (Figure 7), higher to those reported for YSZ and GDC (0.02 and 0.08 S cm^−1^ respectively [2]) and very similar to the best values reported in the literature for LSGM electrolytes synthesized by other routes (see Table 2), which indicates that the mechanochemical method is an excellent technique to obtain homogeneous nanosized LSGM powders. It is worth mentioning that generally, the impurities of SrLaGa_3_O_7_ are isolators and do not conduct, lowering the total conductivity of LSGM. Besides, an excessive sintering and poor densification yield in an excessive grain growth and the presence of pores, which are trapped among the grains or grain boundaries, blocking oxygen ion migration. Both contributions lead to a decrease in the conductivity of LSGM [37,38,39].

For oxygen ionic conductors, the thermal dependence of conductivity is obtained from the well-known Arrhenius plot (Figure 7). An activation energy value of 0.9 eV was found from the slope of the linear fit in the range between 300 and 800 °C. This value is in the same range of activation energy of LSGM synthesized by other synthesis routes.

The impedance spectrum of the LSM//LSGM half-cell is shown in Figure 8. To test the functionality of LSGM as an electrolyte, a measurement was made in an oxidizing atmosphere using LSM as cathode. With this experiment, it is possible to identify the electrolyte-electrode interaction and also the versatility of the electrode using LSGM as electrolyte. The contribution of the electrolyte and the electrode can be appreciated (especially at 600 °C, observing a semicircle in the left part of the corresponding spectrum). The impedance spectrum observed at 600 °C is quite similar to the one obtained in Figure 6. However, there are significant differences among impedance spectra when using platinum and LSM electrodes, particularly in the middle-low frequency region associated with charge transfer process, due to the different nature of both materials. For example, when using LSM it has not been possible to obtain curves below 600 °C. Not only, as temperature increases the high frequency region differs from that shown in Figure 6, i.e., a depressed semicircle is can be now observed instead of the previous curved line behavior. Such interesting behavior may be due to the fact that LSM is used as a material suitable for use as cathodes at temperatures close to 800 °C. Fitting the impedance spectra is now more complicated and was not performed, considering that both LSGM and LSM possesses grain and grain boundary conductions, as well as ionic and electronic conductivities. In any case, the total resistance values, determined by the intersection of the spectra with the abscissa axis, are extremely small, which would indicate that there is a good interaction between electrolyte and cathode, and that both LSGM and LSM are good candidates to act as an electrolyte and cathode, respectively.

## 4. Conclusions

(1)Two components of a SOFC were synthesized by mechanochemistry in a significant short time; cathode (La_0.8_Sr_0.2_MnO_3_) and electrolyte (La_0.8_Sr_0.2_Ga_0.8_Mg_0.2_O_3−δ_), both possessing a perovskite structure with pseudo-cubic symmetry and crystalline domains of nanometric character.(2)Mechanochemistry, which is a scalable process and is also considered cost-effective since no heat input is necessary, is an alternative method to synthesize in a powder form SOFC components based on solid solution oxides with perovskite structure.(3)The synthesized La_0.8_Sr_0.2_Ga_0.8_Mg_0.2_O_3−δ_ electrolyte is chemically and thermally stable up to 800 °C. At 900 °C, two secondary phases start to appear, LaSrGaO_4_ and LaSrGa_3_O_7_; however, the first one disappears at 1400 °C, this being the optimum sintering temperature of the electrolyte.(4)The HRTEM analysis of the LSGM electrolyte after heating at 1400 °C indicates that the symmetry of the structure is kept, and although the nanocrystals have grown, the size of the crystalline domain is still quite small.(5)The La_0.8_Sr_0.2_Ga_0.8_Mg_0.2_O_3−δ_ electrolyte is chemically compatible with the electrode La_0.8_Sr_0.2_MnO_3_ at the operating temperature of 800 °C, and the analysis of the SEM micrographs shows good adhesion at the cathode-electrolyte interface.(6)Obtaining the different SOFC components with materials having the same perovskite structural type and using the same synthesis method can represent an advantage, since both factors should minimize the problems associated with chemical and structural compatibility, leading usually to failure in SOFCs.(7)The complex impedance measurement of LSGM indicates that it is a good candidate to be used as an electrolyte in SOFC, with an Ea value of 0.9 eV. The results of the conductivity measurements of the LSM//LSGM half-cell indicate that LSM sample is a good candidate to be used as a cathode when LSGM acts as electrolyte.

## Figures and Tables

**Figure 1 materials-13-01366-f001:**
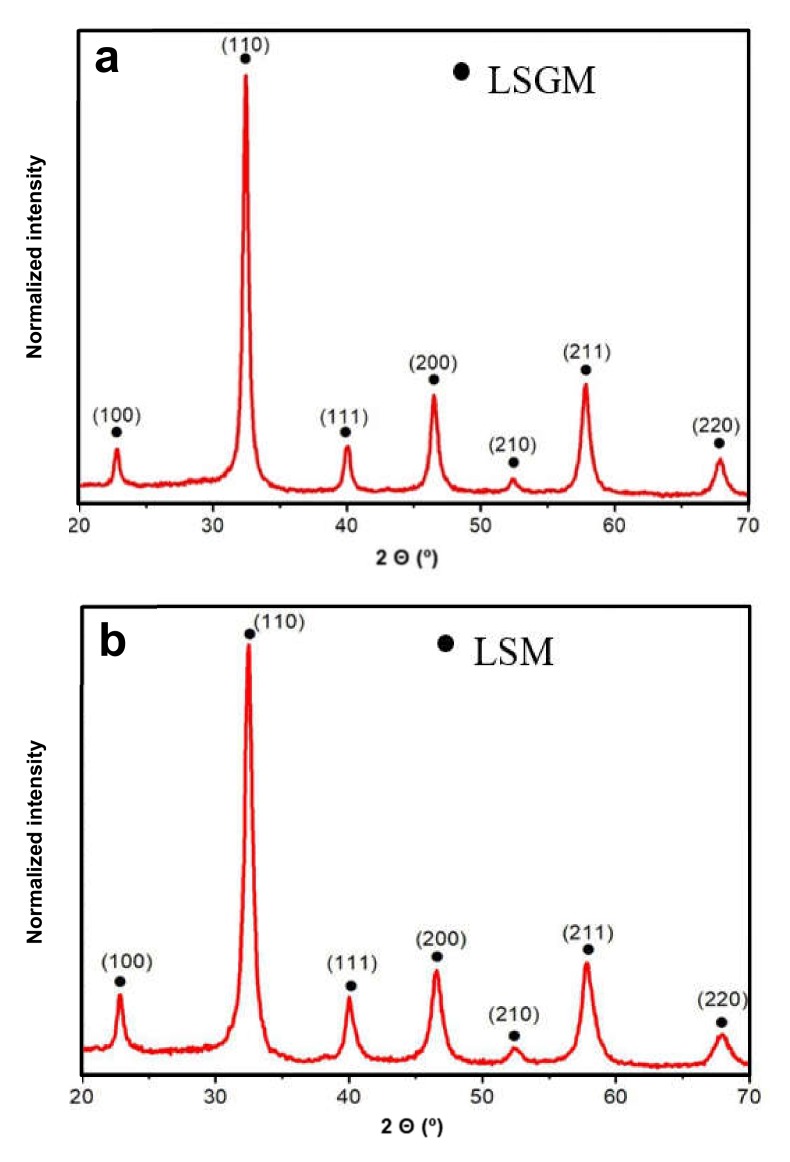
X-ray diffraction patterns of La_0.8_Sr_0.2_Ga_0.8_Mg_0.2_O_3-δ_ (LSGM) (**a**) and La_0.8_Sr_0.2_MnO_3_ (LSM) (**b**) samples synthesized by mechanochemistry, (hkl) indices are marked.

**Figure 2 materials-13-01366-f002:**
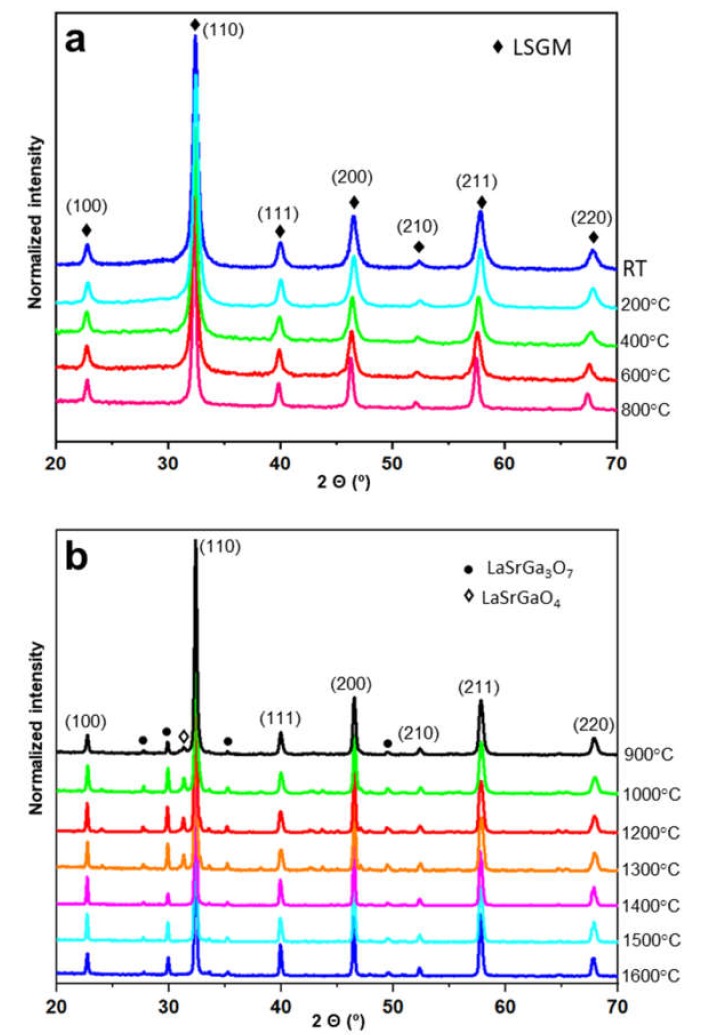
X-ray diffraction patterns of the LSGM sample showing the evolution with the temperature, (**a**) from room temperature (RT) to 800 °C and (**b**) from 900 to 1600 °C.

**Figure 3 materials-13-01366-f003:**
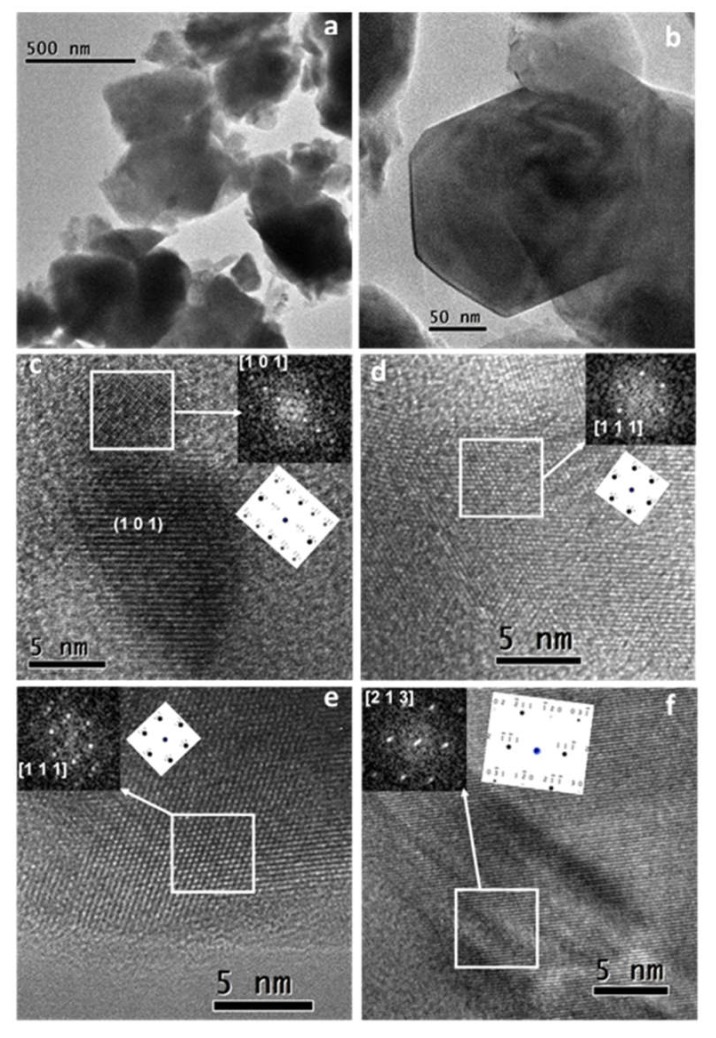
Microstructural results of the LSGM powder sample heated to 1400 °C. (**a**,**b**) TEM images at low magnification, (**c**–**f**) HRTEM micrographs, with some white square areas marked and the corresponding fast Fourier transform (FFT), and simulated EDP inset.

**Figure 4 materials-13-01366-f004:**
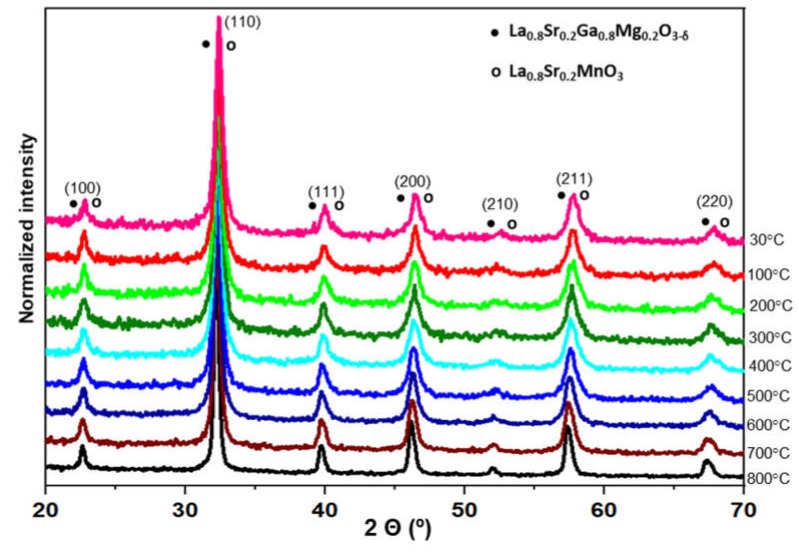
X-ray dynamic patterns of a LSGM/LSM powder mixture (1:1) from 30 to 800 °C.

**Figure 5 materials-13-01366-f005:**
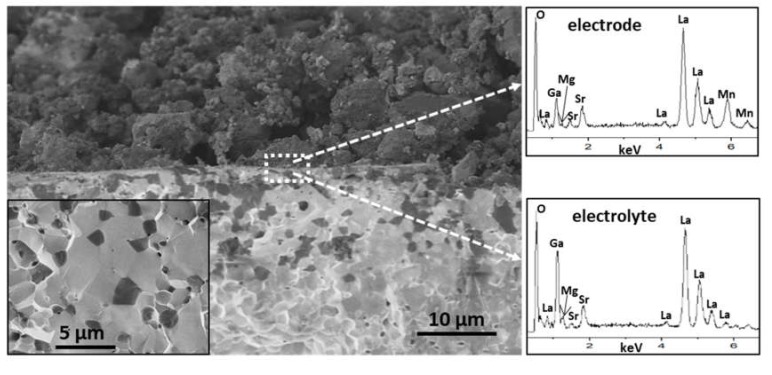
SEM micrograph of the electrode/electrolyte interface cross-section. The inset corresponds to the fracture surface of the electrolyte, from which particle size can be extracted. EDX analyses were taken exactly at the interface from the electrode and electrolyte side, respectively.

**Figure 6 materials-13-01366-f006:**
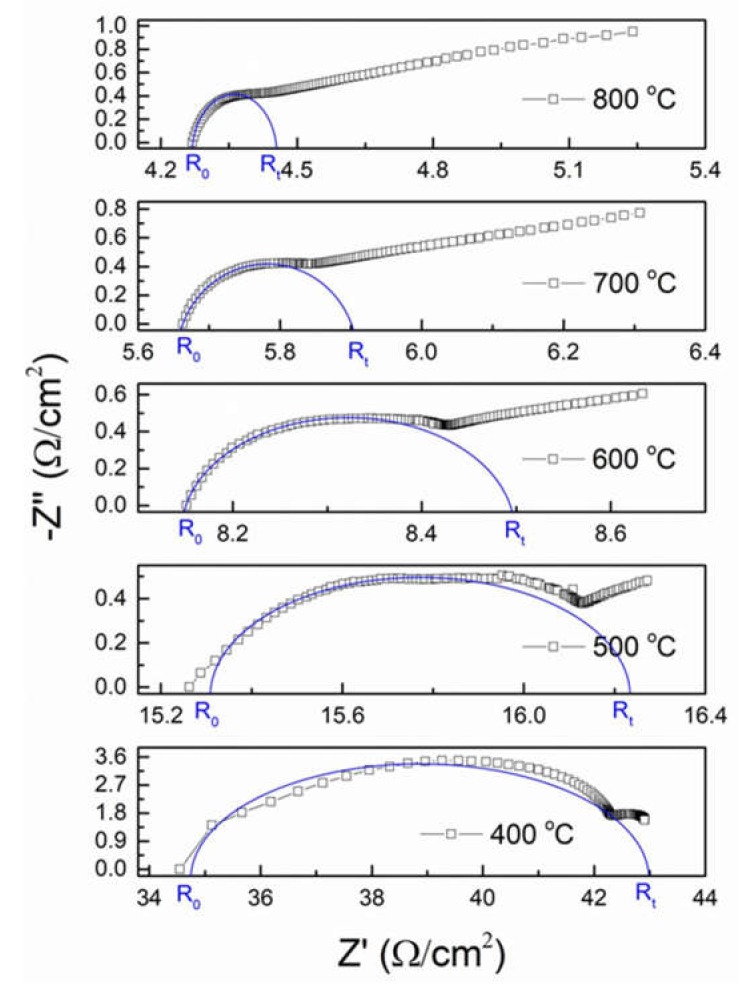
Complex impedance spectrum of the LSGM electrolyte at different temperatures in static air.

**Figure 7 materials-13-01366-f007:**
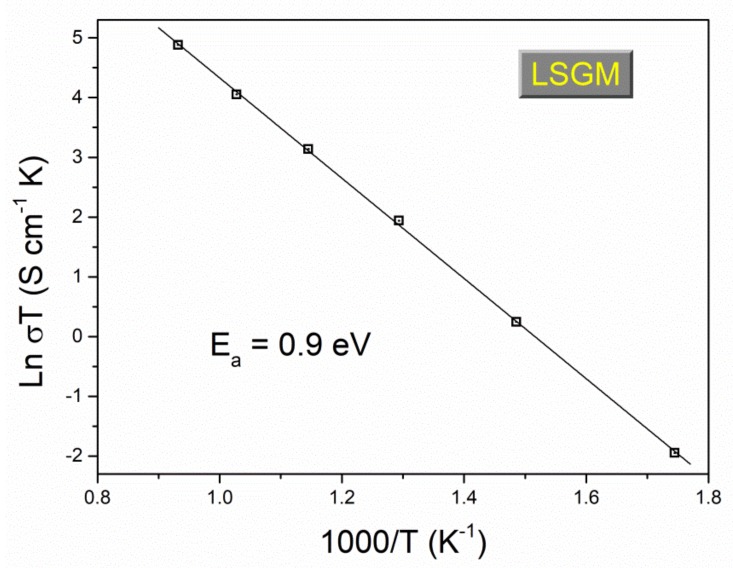
Activation energy obtained from the complex impedance spectrum.

**Figure 8 materials-13-01366-f008:**
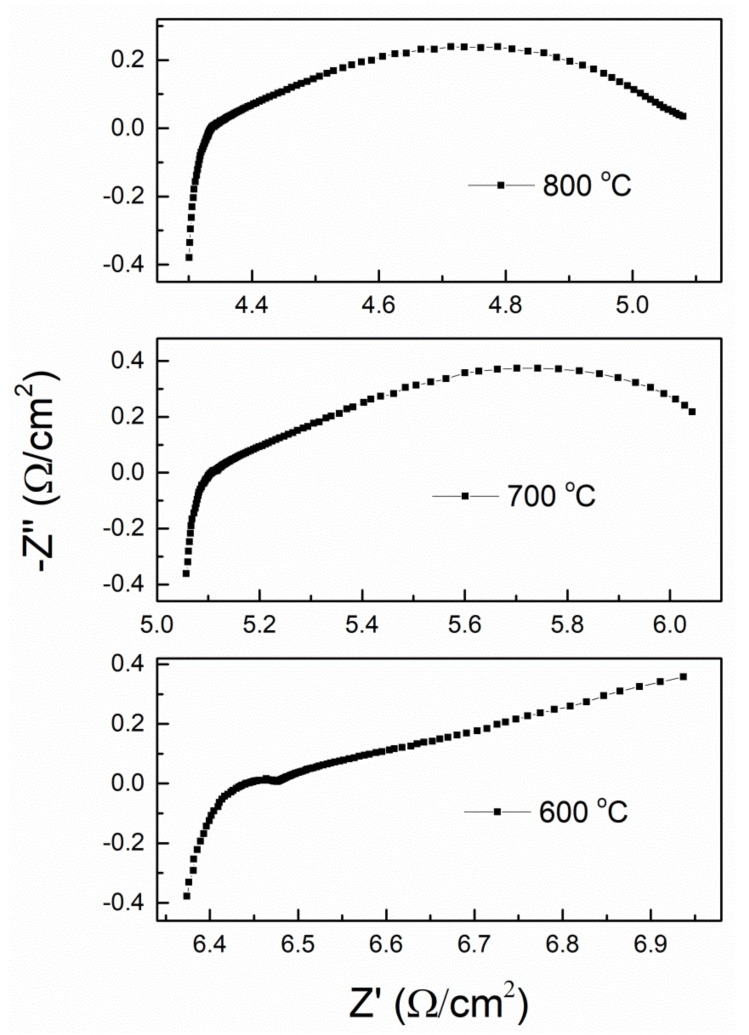
Complex impedance spectra of the LSM//LSGM cell.

**Table 1 materials-13-01366-t001:** Phase percentage in the LSGM samples after heating at several temperatures, quantified by the Rietveld method.

T (°C)	LSMG	LaSrGaO_4_	LaSrGa_3_O_7_	Goodness of Fit
900	89.2	5.2	5.7	1.24
1000	81.2	8.1	10.7	1.32
1200	81.9	6.7	11.4	2.14
1300	81.5	6.3	12.2	1.53
1400	93.7	0	6.3	1.76
1500	89.9	0	10.1	1.78
1600	89.5	0	10.5	1.84

**Table 2 materials-13-01366-t002:** Ionic conductivity at 800 °C of LSGM samples prepared by various synthesis routes.

Synthesis Method	Electrolyte Composition	σ (S cm^−1^)	Ref.
Ethylene glycol	La_0.9_Sr_0.1_Ga_0.8_Mg_0.2_O_2.85_	0.056	[37]
Glycine-nitrate	La_0.8_Sr_0.2_Ga_0.85_Mg_0.15_O_3−_*_δ_*	0.06	[38]
Solid state	La_0.9_Sr_0.1_Ga_0.8_Mg_0.2_O_2.85_	0.03	[39]
Carbonate co-precipitation	La_0.9_Sr_0.1_Ga_0.8_Mg_0.2_O_2.85_	0.045	[40]
Sol-gel method	La_0.9_Sr_0.1_Ga_0.8_Mg_0.2_O_2.85_	0.11	[41]
Hydrothermal urea-precipitation process	La_0.8_Sr_0.2_Ga_0.8_Mg_0.2_O_2.8_	0.056	[42]
Cellulose templating method	La_0.8_Sr_0.2_Ga_0.83_Mg_0.17_O_2.815_	0.042	[43]
Mechanosynthesis	La_0.8_Sr_0.2_Ga_0.8_Mg_0.2_O_3−_*_δ_*	0.093	This work

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
