# Peer review of "Development by Mechanochemistry of La0.8Sr0.2Ga0.8Mg0.2O2.8 Electrolyte for SOFCs"

_materials, 2020, doi:10.3390/ma13061366_

Round 1

Reviewer 1 Report

The manuscript investigates the synthesis, crystal structure, morphology and electrical properties of LSGM and LSM/LSGM synthesized using a mechanochemical process. The authors have synthesized the materials in powder, dense and porous form (depending on each use), conducted XRD, TEM, EDS and EIS on cells to determine the performance and compatibility of the aforementioned materials. The authors show that secondary phases appear on LSGM at temperatures equal to 900C. The LSM/LSGM biphasic electrode is shown to be stable up to 800C without the formation of secondary phases. EIS studies allow the determination of the LSGM ionic conductivity.

The manuscript is interesting and the some of the conclusions are supported by the results. However, there are several areas of improvement before this manuscript can be accepted for publication. As a result, this reviewer believes that major revisions are required. Please find my comments below:

  1. Regarding the term “mechanochemistry”, although this reviewer is aware of it and of its use to synthesize ceramics by reducing the calcination temperature, some of the readers of the manuscript may not be aware of the term. As a result, I would suggest that the authors add 1-2 sentences at the last paragraph of the introduction to inform the reader that materials synthesized by mechanochemistry can be formed without (or with reduced) calcination temperature of the milled precursors. At this point, it is not clear to the reader if calcination is required to form a phase pure material with mechanochemistry or not.
    A statement similar to “we note that no calcination after the milling of the precursors was conducted” could also be added on page 4, line 156.
  1. On page 1, lines 14-15, it is better to denote the oxygen nonstoichiometry of the oxides with δ instead of x and y to avoid confusion with the A and B site doping levels described in the introduction section. I suggest the same nomenclature is used for the rest of the manuscript.
  2. On page 1, line 17, it is better to change “such-as” with “as-prepared”.
  3. On page 2, line 68, please identify what “SLT” stands for.
  4. On page 4, lines 134-135, the authors state the Impedance Spectra were measured at OCV. Given that a real cell operates under an applied voltage, I believe it is reasonable to measure EIS under an applied voltage relevant to the final application. A real fuel/electrolysis cell does not operate under OCV and the application of a voltage could have a direct effect on the material performance and stability.
  5. On page 4, line 137, please specify if atmospheric or synthetic air was used for the measurements. Also, LSM/LSGM seems to have been investigated under air flow, is this the same for LSGM or static air was used here? The same applies for the sintering conditions reported on page 4, line 141.
  6. On page 4, line 146, please specify that the electrolyte disk is LSGM.
  7. Regarding the screen-printing technique, please specify (for both Platinum and LSM/LSMG) the mesh size of the screen and how many passes were made using the squeegee.
  8. Regarding the current collection, what kind or wires and meshes did the authors use (Pt, Au, Cu etc.)? Also, is the setup used for the investigated a single or dual chamber?
  9. On Figure 1, I would suggest the authors plot the XRD patterns using scaled intensity (from zero to one) on the y-axis.
  10. On page 5, lines 173-174, the authors state that calcination of the powder at 1400C makes the LaSrGaO4 secondary phase to disappear. However, if one looks at Figure 2b, the corresponding peaks on both LaSrGa3O7 and LaSrGaO4 phases are still there, even after calcination at 1600C. As a result, I would like the authors to re-plot Figure 2b by zooming into the x-axis at 25-37 degrees to make sure we can see whether the aforementioned peaks are there or not. The plot can be added in the main text or in the supplementary information.
  11. Regarding the formation of the LaSrGa3O7 and LaSrGaO4 phases, are the authors aware of their origin? In other words, why do these phases form at T=900C and why their percentage is reducing at higher temperatures? On page 6, lines 186-187, the authors provide an example from the literature but this example deals with reducing conditions; in this work, the authors are dealing with oxidizing conditions.
  12. On Figures 1 and 2, please add the labels a and b on each corresponding plot.
  13. On Figure 2b, part of the legend is not visible, please correct this.
  14. On table 1, please add another column with the goodness of fit obtained during the Rietveld Refinement of the corresponding XRD patterns.
  15. On page 10, line 238, the authors discuss the densification of the LSGM electrolyte. Please provide an estimate of the relative density of the LSGM electrolyte using, for example, the Archimedes principle. A conclusion based on a SEM image is not representative in terms of porosity/densification.
  16. The quality of Figures 5a and 5b is poor, please correct this.
  17. On Figures 6 and 8, the plots should be Ohms times the area of the electrodes, not just Ohms. This will allow comparison of these values with literature.
  18. On page 11, lines 262-264, the authors should be able to get an estimate of the grain sizes of the LSGM electrolyte by conducting SEM images on the top/bottom and cross section.
  19. On page 12, lines 279-280, the authors state that they have synthesized LSGM free of impurities, but this is not true (see Figure 2 and table 1).
  20. According to table 2, the best ionic conductivity of LSGM is based on the Sol-gel method, but the stoichiometry is not the same as the LSGM electrolyte studied here. Are the authors aware of the conductivity of La0.8Sr0.2Ga0.8Mg0.2O3 synthesized using the Sol-gel method? This would allow for a better and fair comparison of the mechanochemical method shown here with Sol-gel since the same stoichiometry is accounted for. By comparing the La0.9Sr0.1Ga0.8Mg0.2O3 conductivity values, it seems that the Sol-gel method is the best, probably due to lower percentage of secondary phases, but how does the Sol-gel method compare with mechanochemistry for La0.8Sr0.2Ga0.8Mg0.2O3?
  21. How does the ionic conductivity of the LSGM electrolyte synthesized here compare with YSZ or GDC?

Author Response

We would like to thank the reviewer for his critical evaluation of the manuscript and aid in improving on the present draft. Below is an annotated response to each reviewer comment, explaining the corrections and modifications made to clarify the manuscript. Changes in the revised manuscript are highlighted in yellow.

The manuscript investigates the synthesis, crystal structure, morphology and electrical properties of LSGM and LSM/LSGM synthesized using a mechanochemical process. The authors have synthesized the materials in powder, dense and porous form (depending on each use), conducted XRD, TEM, EDS and EIS on cells to determine the performance and compatibility of the aforementioned materials. The authors show that secondary phases appear on LSGM at temperatures equal to 900C. The LSM/LSGM biphasic electrode is shown to be stable up to 800C without the formation of secondary phases. EIS studies allow the determination of the LSGM ionic conductivity.

The manuscript is interesting and the some of the conclusions are supported by the results. However, there are several areas of improvement before this manuscript can be accepted for publication. As a result, this reviewer believes that major revisions are required. Please find my comments below:

  1. Regarding the term “mechanochemistry”, although this reviewer is aware of it and of its use to synthesize ceramics by reducing the calcination temperature, some of the readers of the manuscript may not be aware of the term. As a result, I would suggest that the authors add 1-2 sentences at the last paragraph of the introduction to inform the reader that materials synthesized by mechanochemistry can be formed without (or with reduced) calcination temperature of the milled precursors. At this point, it is not clear to the reader if calcination is required to form a phase pure material with mechanochemistry or not.
    A statement similar to “we note that no calcination after the milling of the precursors was conducted” could also be added on page 4, line 156.

New information about mechanochemistry has been included in the revised version.

  1. On page 1, lines 14-15, it is better to denote the oxygen nonstoichiometry of the oxides with δ instead of x and y to avoid confusion with the A and B site doping levels described in the introduction section. I suggest the same nomenclature is used for the rest of the manuscript.

Corrected in the text.

  1. On page 1, line 17, it is better to change “such-as” with “as-prepared”.

Corrected in the text.

  1. On page 2, line 68, please identify what “SLT” stands for.

Identify in the text.

  1. On page 4, lines 134-135, the authors state the Impedance Spectra were measured at OCV. Given that a real cell operates under an applied voltage, I believe it is reasonable to measure EIS under an applied voltage relevant to the final application. A real fuel/electrolysis cell does not operate under OCV and the application of a voltage could have a direct effect on the material performance and stability.

Impedance spectra were measured at OCV to determine intrinsic properties of the ionic conductor, i.e. its conductivity. For this reason, the application of an external voltage is generally not necessary. Further, the application of an external voltage on an ionic conductor causes a migration of oxide ions and vacancies towards the positive and negative electrodes respectively. This movement of charges produces the polarization of the ionic conductor due to the accumulation of charges on charges of opposite signs and may eventually cause the collapse of the pellet.

In an opposite way, real fuel cells operate under an applied voltage (two separated atmospheres…) and involve more reactions, including electrochemical reactions. Such electrochemical reactions may be promoted or reduced by applying, among others, an external voltage.

  1. On page 4, line 137, please specify if atmospheric or synthetic air was used for the measurements. Also, LSM/LSGM seems to have been investigated under air flow, is this the same for LSGM or static air was used here? The same applies for the sintering conditions reported on page 4, line 141.

Corrected in the text.

  1. On page 4, line 146, please specify that the electrolyte disk is LSGM.

Specified in the text.

  1. Regarding the screen-printing technique, please specify (for both Platinum and LSM/LSMG) the mesh size of the screen and how many passes were made using the squeegee.

Corrected in the text. We applied Pt and LSM by brush painting and apologize for this misunderstanding.

  1. Regarding the current collection, what kind or wires and meshes did the authors use (Pt, Au, Cu etc.)? Also, is the setup used for the investigated a single or dual chamber?

Specified in text.

  1. On Figure 1, I would suggest the authors plot the XRD patterns using scaled intensity (from zero to one) on the y-axis.

Done in the figure.

  1. See Image in document attached.

  • On page 5, lines 173-174, the authors state that calcination of the powder at 1400C makes the LaSrGaO4 secondary phase to disappear. However, if one looks at Figure 2b, the corresponding peaks on both LaSrGa3O7 and LaSrGaO4 phases are still there, even after calcination at 1600C. As a result, I would like the authors to re-plot Figure 2b by zooming into the x-axis at 25-37 degrees to make sure we can see whether the aforementioned peaks are there or not. The plot can be added in the main text or in the supplementary information.

In the figure above, you can see that for 1400, 1500 and 1600 C the XRD patterns only show the LaSrGa3O7 phase. The small peak near the highest peak is an artifact due to the spectral line corresponding to the Tungsten (W) Lα1 line of 1.4767 [Å]. This line becomes gradually visible and more pronounced as the tube gets older. This is because of tungsten from the filament being deposited on the anode. This line is most likely observed next to a strong peak. We have verified this artifact performing new XRD patterns using a new XRD tube and in the revised version, corrected XRD patterns are shown.

  1. Regarding the formation of the LaSrGa3O7 and LaSrGaO4 phases, are the authors aware of their origin? In other words, why do these phases form at T=900C and why their percentage is reducing at higher temperatures? On page 6, lines 186-187, the authors provide an example from the literature but this example deals with reducing conditions; in this work, the authors are dealing with oxidizing conditions.

This is true, but also the formation of these phases depends on their stability with the temperature and not only with the atmosphere conditions. In this sense, in Journal of Power Sources 166 (2007) 35–40 and Solid State Ionics 186 (2011) 44–52, where LSGM was obtained using oxidizing conditions, the presence of the second phases LaSrGa3O7 and LaSrGaO4 was also observed. The authors also used a sintering temperature of 1400 °C to minimize these phases. These references have been included in the revised version.

  1. On Figures 1 and 2, please add the labels a and b on each corresponding plot.

Done in the figure.

  1. On Figure 2b, part of the legend is not visible, please correct this.

Done in the figure.

  1. On table 1, please add another column with the goodness of fit obtained during the Rietveld Refinement of the corresponding XRD patterns.

Added in the table.

  1. On page 10, line 238, the authors discuss the densification of the LSGM electrolyte. Please provide an estimate of the relative density of the LSGM electrolyte using, for example, the Archimedes principle. A conclusion based on a SEM image is not representative in terms of porosity/densification.

The density is measured by the Archimedes principle and the value is written in the text.

  1. The quality of Figures 5a and 5b is poor, please correct this.

A new Figure 5 with a better quality has been included in the revised version.

  1. On Figures 6 and 8, the plots should be Ohms times the area of the electrodes, not just Ohms. This will allow comparison of these values with literature.

Figures 6 and 8 amended.

  1. On page 11, lines 262-264, the authors should be able to get an estimate of the grain sizes of the LSGM electrolyte by conducting SEM images on the top/bottom and cross section.

Grain size was estimated from the fracture surface of the electrolyte (see Fig. 5).

  1. On page 12, lines 279-280, the authors state that they have synthesized LSGM free of impurities, but this is not true (see Figure 2 and table 1).

We wanted to say that the LSGM obtained after milling is free of impurities related with the milling media. The secondary phases evolve during the heating treatment and from our point of view cannot be considered as impurities. In any case, the sentence has been modified in the revised manuscript.

  1. According to table 2, the best ionic conductivity of LSGM is based on the Sol-gel method, but the stoichiometry is not the same as the LSGM electrolyte studied here. Are the authors aware of the conductivity of La0.8Sr0.2Ga0.8Mg0.2O3 synthesized using the Sol-gel method? This would allow for a better and fair comparison of the mechanochemical method shown here with Sol-gel since the same stoichiometry is accounted for. By comparing the La0.9Sr0.1Ga0.8Mg0.2O3 conductivity values, it seems that the Sol-gel method is the best, probably due to lower percentage of secondary phases, but how does the Sol-gel method compare with mechanochemistry for La0.8Sr0.2Ga0.8Mg0.2O3?

We are aware that stoichiometry can affect conductivity in LSGM, but our first results using mechanochemistry are promising. We are currently conducting a new study, once the mechanochemical process optimized, in which we are characterizing powders with different stoichiometries.In parallel, we are also searching for other more references to compare our results with samples that really match the same stoichiometry. We have not found a reference in literature reporting the conductivity of La0.8Sr0.2Ga0.8Mg0.2O3 prepared by conventional sol-gel method. This is why we included a similar LSGM sample and not the same one.

  1. How does the ionic conductivity of the LSGM electrolyte synthesized here compare with YSZ or GDC?

Ionic conductivities for YSZ and GDC have been added.

Reviewer 2 Report

The authors report on the synthesis of LSGM by mechanosynthesis followed by sintering at 1400°C and combine this electrolyte with LSM cathode also prepared by mechanosynthesis (ref. 29 and 30). This combination of materials is not new (Ishihara et al. classical paper in JACS on gallate electrolytes, cited as ref 9, dates back to 1994) and cannot be considered as a hot topic. However, the reported study could be of moderate interest for readers considering the suitability of mechanosynthesis for these materials.

There are several points that should be addressed before the work can be considered for publication:

  • The abstract should be modified to make it clear that the originality of the work lies with the synthesis method, not the choice of materials. In addition, the LSM/LSGM assembly is not a complete cell and should not be called a "sofc". The issue of the compatibility of LSGM with an anode material should be discussed in the introduction.
  • Line 155: patterns at increasing milling time, mentioned in the text, should be shown in Figure 1. SEM micrographs would be a valuable addition to provide information on grain size distribution as a function of milling time and allow for comparison with other works.
  • Please check all space group symbols as there are some typos, e.g. line 158 and 203 : space group 221 is Pm-3m, not Pm3m ; Line 172 space group I4/mmm
  • The appearance of LaSrGaO4 and LaSrGa3O7 at intermediate temperatures suggests the presence of amorphous material in the milled precursor. Is there any evidence for this in the evolution of the intensity (= peak area) of the XRD patterns?
  • In figure 2a, there seems to be a significant shift of the reflections towards smaller 2theta values at 800°C, is this intrinsic to the sample or an error in sample height in the diffractometer? The authors should plot the evolution of the cell parameter vs. temperature.
  • The mixing of colors in the EDX map of Fig 5b does not allow to assess clearly the element distribution. Maps of individual elements are needed, with a scale bar. More quantitative information (such as a line scan across the interface) would be desirable.
  • Even if the authors were not able to fit the impedance spectra with an equivalent circuit, they should at least mark in the plots the special points (intercepts, etc) mentioned in the discussion.
  • Conclusion #2: the paper does not provide data on the scalability and cost of the synthesis route, so no claims should be made in the conclusion

Author Response

We would like to thank the reviewer for his critical evaluation of the manuscript and aid in improving on the present draft. Below is an annotated response to each reviewer comment, explaining the corrections and modifications made to clarify the manuscript. Changes in the revised manuscript are highlighted in yellow.

The authors report on the synthesis of LSGM by mechanosynthesis followed by sintering at 1400°C and combine this electrolyte with LSM cathode also prepared by mechanosynthesis (ref. 29 and 30). This combination of materials is not new (Ishihara et al. classical paper in JACS on gallate electrolytes, cited as ref 9, dates back to 1994) and cannot be considered as a hot topic. However, the reported study could be of moderate interest for readers considering the suitability of mechanosynthesis for these materials.

There are several points that should be addressed before the work can be considered for publication:

  • The abstract should be modified to make it clear that the originality of the work lies with the synthesis method, not the choice of materials. In addition, the LSM/LSGM assembly is not a complete cell and should not be called a "sofc". The issue of the compatibility of LSGM with an anode material should be discussed in the introduction.

The abstract has been modified. The LSM/LSGM assembly has been called half-cell throughout the text. Regarding the compatibility of LSGM with the electrodes, this is a general condition for any material used as an electrolyte and it was indicated in the first paragraph of the introduction.

  • Line 155: patterns at increasing milling time, mentioned in the text, should be shown in Figure 1. SEM micrographs would be a valuable addition to provide information on grain size distribution as a function of milling time and allow for comparison with other works.

We have tried to perform this type of characterization at increasing milling time, but we have had many problems associated with the presence of SrO and re-hydration and re-carbonation processes. The results were influenced by them and they did not allow the results to be of quality.

  • Please check all space group symbols as there are some typos, e.g. line 158 and 203 : space group 221 is Pm-3m, not Pm3m ; Line 172 space group I4/mmm.

It has been corrected in the text.

  • The appearance of LaSrGaO4 and LaSrGa3O7 at intermediate temperatures suggests the presence of amorphous material in the milled precursor. Is there any evidence for this in the evolution of the intensity (= peak area) of the XRD patterns?

We have not observed evidence of amorphous phase in the XRD patterns. As mentioned in the manuscript (lines 200-212), these secondary phases are frequently observed when LSGM is obtained from different processes.

  • In figure 2a, there seems to be a significant shift of the reflections towards smaller 2theta values at 800°C, is this intrinsic to the sample or an error in sample height in the diffractometer? The authors should plot the evolution of the cell parameter vs. temperature.

We have checked that this shift was an error in sample height and it has been corrected in the revised version. We did not observe significant deviations in cell parameters with temperature.

  • The mixing of colors in the EDX map of Fig 5b does not allow to assess clearly the element distribution. Maps of individual elements are needed, with a scale bar. More quantitative information (such as a line scan across the interface) would be desirable.

We agree that the quality of Figure 5 was not good and it has been modified following the suggestion of another of the reviewers.

  • Even if the authors were not able to fit the impedance spectra with an equivalent circuit, they should at least mark in the plots the special points (intercepts, etc) mentioned in the discussion.

Intercepts have been marked in Figure 6.

  • Conclusion #2: the paper does not provide data on the scalability and cost of the synthesis route, so no claims should be made in the conclusion

This conclusion has been rewritten in the revised version.

Reviewer 3 Report

Fuel cell is an alternative clean energy source that produces electrical energy and some heat through the electrochemical reaction between the gaseous fuels, such as hydrogen, and an oxidant. SOFCs have attracted significant attention due to their prospective applications in electrochemical devices. However, development of this technology has been bottlenecked by its high operating temperature of over 800 °C, which limits the choice of materials and causes the rapid degradation rate of the system components. Over the last few years, research has focused on lowering the SOFC operating temperature well below 750 °C, thereby lowering the system cost, improving the long-term stability, and shortening start-up time. The performance of low-to-intermediate temperature (400 – 800 °C) SOFCs depends on the properties of electrolytes used. To address this issue, authors have developed mixed oxides possessing perovskite structure. The presented work providing material synthesis while replacing La by Sr and Ga by Mg, and characterization is significant and worth to publish. However, the work in its present form is not publishable and need some revisions before rendering a final decision. 

My specific points are below

  • Abstract Lines 20 – 23 is confusing though. stable up to 800 °C and from 900 °C the secondary phases appear – OK. Then the next sentence “optimum ….” Is contradicting.
  • Originality of the work need to be emphasised in the last few lines of the introduction, the current version is very weak.
  • In the introduction; electrochemical reaction occurring at the cathode and anode for SOFC need to be provided.
  • Please provide the ionic radius for the dopants.
  • Line 115 – please provide the equation used for Debye Scherrer calculation.
  • Experimental: Line 119 – How frequently the XRD measurements were made?
  • Are the impedance spectra in-situ or ex-situ?
  • Figure 1; For XRD; please provide the lattice parameters.
  • Results and Discussion Line 175; “The XRD peak broadening with the influence of temperature and doping” has been well explained in the literature (Electrochem Solid State Lett. 2011, 14(6), A86-A89) please refer.
  • Line 188; sentence need to be finished like “Sr cation dopant”
  • Line 204: what is FFT?
  • The shapes obtained for the material through TEM analysis need to be well discussed.
  • 3e; are those contrast seen refers to crystalline structure fringes?
  • Section 3.2: Line 216 “affecting negatively” does this mean reducing the conductivity? Please check.
  • Section 3.2: The morphology of the electrode with a porous nanostructure can be discussed using the literature (such as Progress in Solid State Chem. 40 (2012) 1-5) discussed for electrode-electrolyte interface cathode.
  • Section 3.3: What does the semicircle corresponds to grain/grain-boundary?
  • The impedance spectra and their resistance with reference to the intercepts need to be appropriately referenced back to the similar work.
  • May be a good idea to keep the y-axes values identical for a comparison.
  • In all XRD figures, x-axis units need to be provided like (2 theta/degrees) / Cu K- alpha.
  • How the LSGM compare with gadolinium-doped-ceria?
  • Conclusion No. 6 is unclear.

Author Response

We would like to thank the reviewer for his critical evaluation of the manuscript and aid in improving on the present draft. Below is an annotated response to each reviewer comment, explaining the corrections and modifications made to clarify the manuscript. Changes in the revised manuscript are highlighted in yellow.

Fuel cell is an alternative clean energy source that produces electrical energy and some heat through the electrochemical reaction between the gaseous fuels, such as hydrogen, and an oxidant. SOFCs have attracted significant attention due to their prospective applications in electrochemical devices. However, development of this technology has been bottlenecked by its high operating temperature of over 800 °C, which limits the choice of materials and causes the rapid degradation rate of the system components. Over the last few years, research has focused on lowering the SOFC operating temperature well below 750 °C, thereby lowering the system cost, improving the long-term stability, and shortening start-up time. The performance of low-to-intermediate temperature (400 – 800 °C) SOFCs depends on the properties of electrolytes used. To address this issue, authors have developed mixed oxides possessing perovskite structure. The presented work providing material synthesis while replacing La by Sr and Ga by Mg, and characterization is significant and worth to publish. However, the work in its present form is not publishable and need some revisions before rendering a final decision. 

My specific points are below

  • Abstract Lines 20 – 23 is confusing though. stable up to 800 °C and from 900 °C the secondary phases appear – OK. Then the next sentence “optimum ….” Is contradicting.

It has been changed in the abstract. We wanted to say that 1400 °C was the best temperature to sinterize the electrolyte because LaSrGaO4 phase disappears and the percentage of LaSrGa3O7 phase was very small.

  • Originality of the work need to be emphasised in the last few lines of the introduction, the current version is very weak.

The last paragraph of the introduction has been modified according to reviewer suggestion.

  • In the introduction; electrochemical reaction occurring at the cathode and anode for SOFC need to be provided.

The electrochemical reactions are well known for the scientific community and we think it is not necessary to be written.

See equation in document attached

The cathode catalyzes the oxygen reduction reaction:

See equation in document attached

The anode catalyzes the oxidation of fuel, for example, for hydrogen:

Besides, this information can be obtained, for example, in references 1 and 2 that are review papers on SOFCs materials and technology.

  • Please provide the ionic radius for the dopants.

This information is provided in the revised version.

  • Line 115 – please provide the equation used for Debye Scherrer calculation.

The experimental procedure concerning XRD analyses have been modified and Scherrer equation was not employed in the revised version.

  • Experimental: Line 119 – How frequently the XRD measurements were made?

As it is indicated in the manuscript (lines 126-128), the HT-XRD patterns were recorded at atmospheric pressure at temperature intervals of 50 °C from room temperature to 800 °C.

  • Are the impedance spectra in-situ or ex-situ?

To be honest with the reviewer, I do not really understand the question and hope to provide a proper answer, which is as followed: For impendance measurements, LSGM powder is obtained by directly milling the oxide powders at high energy. Then, the LSGM pellet is obtained by first compacting the LSGM powder using a uniaxial press (“green body”) to finally sinter  at 1400 °C. Once we obtain the final pellet, we apply Pt at both sides to get good electrical contacts and measure its impedance. During EIS measurement, the whole pellet is exposed to airflow and kept to the temperatures mentioned in the text.   

  • Figure 1; For XRD; please provide the lattice parameters.

Lattice parameters were indicated in the text: a=0.390 nm for LSGM (line 168) and a=0.391 nm for LSM (line 174).

  • Results and Discussion Line 175; “The XRD peak broadening with the influence of temperature and doping” has been well explained in the literature (Electrochem Solid State Lett. 2011, 14(6), A86-A89) please refer.

The reference has been added.

  • Line 188; sentence need to be finished like “Sr cation dopant”

It was modified.

  • Line 204: what is FFT?

It is indicated in line 136: fast Fourier transform.

  • The shapes obtained for the material through TEM analysis need to be well discussed.

It is well known that when small particles with cubic structure grow with temperature, they can develop different polyhedral shapes (see for example S. Onaka, Philosophical Magazine Letters, 86, No. 3 (2006) 175–183)

  • 3e; are those contrast seen refers to crystalline structure fringes?

The black-white-grey contrast refers to the atomic structure order in the related orientation. It can be structure fringes or atomic columns if the region is well oriented in a zone axis of the structure (white square marked).

  • Section 3.2: Line 216 “affecting negatively” does this mean reducing the conductivity? Please check.

Yes, this has been clarified in the text.

  • Section 3.2: The morphology of the electrode with a porous nanostructure can be discussed using the literature (such as Progress in Solid State Chem. 40 (2012) 1-5) discussed for electrode-electrolyte interface cathode.

We have carefully read the reference you suggest; however, in our case, the porosity is at a microstructural level, not in the nanostructure. The electrode is painted on the electrolyte, with a slurry formed by very small grains that are further heated up to 800 °C to produce the adhesion to the electrolyte surface, and the grain size is maintained about 0.2 µm with a microporous structure.

  • Section 3.3: What does the semicircle corresponds to grain/grain-boundary?

As mentioned in the text, it has not been possible to discern the partial contribution of grains and grain boundaries. Therefore, the semicircle corresponds to the total ionic impedance, i.e. both grain plus grain boundary contributions.

  • The impedance spectra and their resistance with reference to the intercepts need to be appropriately referenced back to the similar work.

Intercepts have been added in Figure 6.

  • May be a good idea to keep the y-axes values identical for a comparison.

The Figures have been corrected.

  • In all XRD figures, x-axis units need to be provided like (2 theta/degrees) / Cu K- alpha.

Done in the figures

  • How the LSGM compare with gadolinium-doped-ceria?

Ionic conductivities for YSZ and GDC have been added.

  • Conclusion No. 6 is unclear.

It has been rewritten

Round 2

Reviewer 1 Report

The authors have addressed most of my comments and they have revised their manuscript considerably. Overall, the quality of the manuscript has been improved significantly. However. before allowing for the manuscript to be accepted for archival publication, there are 3 last comments that I would like the authors to address:

  1. Table 1 shows the quantification of the different phases within LSGM based on Rietveld refinement using the corresponding XRD patterns at different temperatures. The closer the goodness of fit to the value of one, the better the fitting of the model to the XRD pattern and hence, the more accurate the phase quantification. In this reviewer’s knowledge, a goodness of fit less than 2 should be acceptable. In this case, however, the goodness of fit is between 2.33-6.39. Especially at T higher than 1300C, the goodness of fit exhibits values in the range 3-5. This shows that the model used for the Rietveld refinement does not fit the XRD patterns with good accuracy; I would characterize the fitting of moderate accuracy. Such values of goodness of fit reveal a poor fitting. As a result, I believe that the phase percentages shown in table 1 are not very accurate. Can the authors provide an error bad on the estimated phase percentages due to the aforementioned relatively poor fitting of the XRD patterns?
  2. On figures 6 and 8, the authors should plot the impedance times the active surface area, i.e. Ohms multiplied by cm^2. Figure 6 plots Ohms divided by cm^2 while figure 8 plots Ohms.
  3. There are English grammatical errors in the manuscript that the authors should correct before publication.

Author Response

We would like to thank the reviewer for his critical evaluation of the manuscript and aid in improving on the present draft. Below is an annotated response to each reviewer comment, explaining the corrections and modifications made to clarify the manuscript. Changes in the revised manuscript are highlighted in yellow.

The authors have addressed most of my comments and they have revised their manuscript considerably. Overall, the quality of the manuscript has been improved significantly. However. before allowing for the manuscript to be accepted for archival publication, there are 3 last comments that I would like the authors to address:

  1. Table 1 shows the quantification of the different phases within LSGM based on Rietveld refinement using the corresponding XRD patterns at different temperatures. The closer the goodness of fit to the value of one, the better the fitting of the model to the XRD pattern and hence, the more accurate the phase quantification. In this reviewer’s knowledge, a goodness of fit less than 2 should be acceptable. In this case, however, the goodness of fit is between 2.33-6.39. Especially at T higher than 1300C, the goodness of fit exhibits values in the range 3-5. This shows that the model used for the Rietveld refinement does not fit the XRD patterns with good accuracy; I would characterize the fitting of moderate accuracy. Such values of goodness of fit reveal a poor fitting. As a result, I believe that the phase percentages shown in table 1 are not very accurate. Can the authors provide an error bad on the estimated phase percentages due to the aforementioned relatively poor fitting of the XRD patterns?

A more careful Rietveld analysis was performed, which allowed obtaining better figures of merit. The new quantification results in the revised version have shown a slightly higher content for the secondary phases, but the same trend was observed, so the conclusions are not modified.

  1. On figures 6 and 8, the authors should plot the impedance times the active surface area, i.e. Ohms multiplied by cm2. Figure 6 plots Ohms divided by cm2 while figure 8 plots Ohms.

Figure 8 was corrected in the revised version.

  1. There are English grammatical errors in the manuscript that the authors should correct before publication.

A careful English editing was made and several errors were corrected.

Reviewer 3 Report

Authors have addressed my queries raised earlier and to this reviewer's opinion this version is suitable to publish.

Author Response

We would like to thank again the reviewer for his critical evaluation of the manuscript and aid in improving on the present draft.